# Insights into the Antimicrobial Activity of Hydrated Cobaltmolybdate Doped with Copper

**DOI:** 10.3390/molecules26051267

**Published:** 2021-02-26

**Authors:** Layane A. L. Silva, André A. L. Silva, Maria A. S. Rios, Manoel P. Brito, Alyne R. Araújo, Durcilene A. Silva, Ramón R. Peña-Garcia, Edson C. Silva-Filho, Janildo L. Magalhães, José M. E. Matos, Josy A. Osajima, Eduardo R. Triboni

**Affiliations:** 1Interdisciplinary Laboratory Advanced Materials, Federal University of Piauí, Teresina 64049-550, Brazil; layanealmeida@ufpi.edu.br (L.A.L.S.); rraudelp@gmail.com (R.R.P.-G.); edsonfilho@ufpi.edu.br (E.C.S.-F.); jmematos@ufpi.edu.br (J.M.E.M.); 2Supramolecular Self-Assembly Laboratory, Federal University of Piauí, Teresina 64049-550, Brazil; andrelima@ufpi.edu.br (A.A.L.S.); janildo@ufpi.edu.br (J.L.M.); 3Group of Technological Innovations and Chemical Specialties, Federal University of Ceará, Fortaleza 60455-760, Brazil; alexsandrarios@ufc.br; 4Biodiversity and Biotechnology Research Center, Federal University of Delta of Parnaíba, Parnaíba 64202-020, Brazil; manoelbrito93@gmail.com (M.P.B.); alyne_biomed@hotmail.com (A.R.A.); durcileneas@gmail.com (D.A.S.); 5Academic Unit of Cabo de Santo Agostinho, Federal Rural University of Pernambuco, Cabo de Santo Agostinho 52171-900, Brazil; 6Nanotechnology and Process Engineering-NEP, University of São Paulo, Lorena 12602-810, Brazil

**Keywords:** co-precipitation, *Staphylococcus aureus*, *Escherichia coli*

## Abstract

Molybdates are biocidal materials that can be useful in coating surfaces that are susceptible to contamination and the spread of microorganisms. The aim of this work was to investigate the effects of copper doping of hydrated cobalt molybdate, synthesized by the co-precipitation method, on its antibacterial activity and to elucidate the structural and morphological changes caused by the dopant in the material. The synthesized materials were characterized by PXRD, Fourier Transformed Infrared (FTIR), thermogravimetric analysis/differential scanning calorimetry (TG/DSC), and SEM-Energy Dispersive Spectroscopy (SEM-EDS). The antibacterial response of the materials was verified using the Minimum Inhibitory Concentration (MIC) employing the broth microdilution method. The size of the CoMoO_4_·1.03H_2_O microparticles gradually increased as the percentage of copper increased, decreasing the energy that is needed to promote the transition from the hydrated to the beta phase and changing the color of material. CoMoO_4_·1.03H_2_O obtained better bactericidal performance against the tested strains of *Staphylococcus aureus* (gram-positive) than *Escherichia coli* (gram-negative). However, an interesting point was that the use of copper as a doping agent for hydrated cobalt molybdate caused an increase of MIC value in the presence of *E. coli* and *S. aureus* strains. The study demonstrates the need for caution in the use of copper as a doping material in biocidal matrices, such as cobalt molybdate.

## 1. Introduction

The current pandemic situation that has been caused by Coronavirus Disease 2019 (COVID-19) has increased investigation of biocidal materials that can be used to coat surfaces, which act as indirect routes of infection and spread of microorganisms, such as the coronavirus of severe acute respiratory syndrome 2 (SARS-CoV-2) [1,2,3]. Recently, studies demonstrated the feasibility of materials containing Mo in its formulation with high antimicrobial activity [4,5], called molybdates, which have an important role in the structure and function of metal or proteins in certain biological systems [6,7,8]. Articles have recently been published on the antimicrobial properties of silver molybdates [9], lead [10], copper [2], zirconium [11], titanium [12], iron [13], and zinc [14].

Cobalt molybdate (CoMoO_4_) is a crystalline semiconductor that can exist in the α-CoMoO_4_, β-CoMoO_4_, hp-CoMoO_4_, and hydrated (CoMoO_4_·nH_2_O) phases, presenting magnetic [15], capacitive [16], catalytic properties [17], and photocatalytic [18] properties, which give it a wide range of applications. However, few studies have explored its antimicrobial potential. Amanulla et al. [19] reported the synthesis of β-CoMoO_4_ nanorods via co-precipitation, being used in the formulation of a nanocomposite (β-CoMoO_4_-Co_3_O_4_), whose antibacterial activity was found for gram-negative (*Pseudomonas aeruginosa* and *Escherichia coli*) and gram-positive (*Staphylococcus aureus*) bacteria while using the agar diffusion method; Meng and Xiong [20] observed that hydrothermally obtained micrometric rods of H-CoMoO_4_ that have bactericidal behavior against *E. coli* and *S. aureus*. Kong et al. [6] evidenced that α-CoMoO_4_ nanorods, which were synthesized by the microemulsion method, formed an inhibition halo against *E. coli* with a diameter of approximately 2.5 cm.

Several synthetic routes have already been studied to obtain cobalt molybdate, such as hydrothermal [21], microemulsion [22], sonication [23], and sol-gel [24]. However, the co-precipitation route is simpler as well as economical, ecological, and energetically advantageous, in addition to being easily scalable for industrial production, which is based on the mixture of precursor salts in aqueous solution, followed by stoichiometric precipitation [25,26]. Precursors, temperature, pH, \order of addition of reagents, ionic strength of the medium, molar ratio of forming ions, agitation speed, and other parameters considerably affect the nature, homogeneity, size, morphology, surface area, and degree of long-range organization in the crystalline lattices of the obtained particles [17,27,28].

In addition to the methodology that was used in the synthesis process, another resource widely employed to modify the properties of semiconductors is doping, whose effects are related to the introduction of defects in their respective crystalline lattices. Calcium molybdates [29], zinc [30], and nickel [31] were doped with cobalt, for application in humidity sensors and photoluminescence studies. Tantraviwat et al. [32] studied doping with tungsten for application in the reaction of evolution of oxygen. Costa et al. [25] verified the effect of doping with europium on the phase transition of the CoMoO_4_, α and β polymorphs.

Doping is an important strategy for changing the antibacterial activity of semiconductor materials, which also varies according to the type of bacteria (gram-negative and gram-positive). Wang et al. [33], for example, demonstrated that the doping of MoS_2_ by copper ions improves the antibacterial activity against gram-positive strains of *S. aureus*. Okeke et al. [34] observed that gram-negative bacteria (*E. coli* and *P. aeruginosa*) were more sensitive to ZnO nanoparticles doped with 3 and 5% Cu. Meanwhile, Zhang et al. [35] found that doping TiO_2_ with 6% Cu improved its antibacterial properties against *S. aureus* and *E. coli* bacteria.

On the other hand, whether the doping process can endow semiconductor materials with biocidal effects is controversial. For example, in the work conducted by Yang et al. [36], the Er-ZnO/SiO_2_ material exhibited good antibacterial activity against *E. coli* and *S. aureus.* However, the antibacterial properties of the Er-ZnO/SiO_2_ material depended on the amount that the Er^3+^ concentration was increased, which, at certain concentrations, decreased the antibacterial activity when the molar ratio from Zn to Er increased. Aftab et al. [37] reported that pure NiO is a very good antibacterial agent against *E. coli* bacteria, but its antibacterial capacity gradually decreases when doped with 0.1–25% Cu. Khan et al. [38] showed that ZnO that was doped with 1 and 5% Cu exhibited antibacterial activity only against gram-positive bacteria (*B. subtilis* and *S. aureus*).

The objective of this work was to investigate the effects of copper doping of hydrated cobalt molybdate, synthesized by the co-precipitation method, in order to elucidate the structural and morphological changes that are caused by the dopant in the matrix lattice of the material and study its antibacterial activity against *Staphylococcus aureus* and *Escherichia coli*.

## 2. Results and Discussion

### 2.1. Structure of Materials

X-ray diffraction analyses were performed in all synthesized samples to verify the phase formation (Figure 1). As noted, the XRD patterns are similar and they were indexed according to the reference code (ICDD: 153169), (Figure 1a–d). The highest intensity peak in the diffraction profile of hydrated cobalt molybdate without doping (Figure 1a) is related to the family of crystalline (003) plane, located around 29° in 2θ. Similar results were obtained by Costa [25] in the synthesis of hydrated cobalt molybdate phase by co-precipitation method by Eda et al. [39] and Liu et al. [40] while using the hydrothermal route. For the doped samples (Figure 1b–d), the XRD diffractograms showed a change in the relative intensities of the diffraction peaks.

The progressive increase in the intensity of the peaks is more evident for the family of (001) and (100) planes, in relation to (003) plane. This result is interesting, since it could be a clear effect of Cu inclusion in the structure, causing intrinsic defects, such as oxygen vacancies and stress. Structural defects and copper insertion in the hydrated cobalt molybdate can modify the nucleation mechanisms during phase formation, which could cause changes in crystallographic orientation. It is known that nanostructures with different sizes and/or shapes can produce more complex XRD patterns, making structural analysis more difficult. In this study, the Rietveld refinement method was used to determine the lattice parameters and be able to evaluate the effects of Cu concentration on the studied structure. Figure 1 also shows the calculated pattern of XRD (red line) and the difference profile (blue line) for all samples, which were obtained by using the Rietveld analysis. These parameters vary as the Cu concentration increases, according to the obtained values for the lattice constants (*a*, *b*, and *c*) (Table 1).

The continuous increase in lattice constant *a*, when the dopant concentration varies from 0.00 to 0.12, suggests that all Cu (larger ions) replace the Co (smaller ions) in the hydrated cobalt molybdate structure, because the ionic radii for Co and Cu are 0.58 Å and 0.73 Å, respectively [41]. On the other hand, the lattice parameters *b* and *c* vary and they depend on the Cu content in the structure. These changes are caused by the structural defects and internal stress that can be induced due to the addition of the dopant, since the dopant ions inside the structure of the materials can provoke substitutional defects, vacancies, and interstitial defects. Similar results were reported by Costa et al. [25] for β− and α−CoMoO doped with Eu.

### 2.2. Spectroscopic Analysis

The spectrum of CoMoO_4_·nH_2_O (Figure 2a) and Co_1−x_Cu_x_MoO_4_·nH_2_O (Figure 2b–d) have strong absorption bands in the region between 1000 and 400 cm^−1^. The typical stretch vibration υ_3_(Mo–O) that s related to distorted tetrahedral clusters (MoO_4_) can be observed at 906 and 739 cm^−1^ in the same way that the asymmetric stretching mode of two units of this same cluster was observable at 904 cm^−1^ [42,43,44,45].

The two bands with the maximum observed at 813 and 851 cm^−1^ refer to the stretching vibration of the group (Mo–O–Mo) centered on the oxygen atom at the edge common to two clusters of (MoO_4_)_Td_ [26]. The decrease in the relative intensity of the narrow band with the maximum located at 432 cm^−1^ is due to the increased amount of dopant. This band is due to the superposition of the active modes ʋ_4_ and ʋ_5_ of MoO and ʋ_3_ of the octahedral cluster of CoO_6_ in the formation of CoMoO_4_ groups [26,44,45]. This may indicate that doping with copper results in a loss of coordination water, since, in the hydrated phases of CoMoO_4_, the water molecule is linked to Co^2+^ in clusters of (CoO_6_)_Oh_, where the replacement of cobalt ions by copper ions is expected. In addition, bands that are found in the region between 1737 and 1641 cm^−1^ correspond to the stabilization mode of the O–H connection of the coordination and crystallization waters. The two weak bands at 1612 and 1425 cm^−1^ are attributed to the symmetrical stretching and asymmetric C-O ionized carboxylate from CO_2_ that is adsorbed on the surface of the respective materials [19,46].

### 2.3. Thermal Analysis

The thermal gravimetric analysis (TG) curve that was presented by the Co_1−x_Cu_x_MoO_4_∙nH_2_O materials (Figure 3a–d) showed that the sample dehydration process occurred in two stages. The first event was the elimination of water molecules adsorbed in the cavities of the respective structures was observed. In the second event, the two overlapping peaks in the DTG curves (Figure 3a–d) characterize the transition process from the hydrated phase to the β phase, which occurs after the elimination of the water, through an exothermic event that was observed in the DTA curves (Figure 3e–h).

Other studies have also reported the dehydration procedure for cobalt molybdate, obtained via co-precipitation, which occurred through the release of two different types of water molecules: (i) units that are physically adsorbed in the crystalline structure cavities, eliminated in the first event and (ii) structural units that are connected to the lattice of the respective material, removed in the second event [15,27]. Kim et al. [47] reported that hydrated cobalt molybdate, also obtained via co-precipitation, exhibited a gradual loss of mass around 100 to 400 °C, which represented the removal of water molecules that were connected to the sample lattice, remaining at the end of the heating process ~87.5% of the residual sample of anhydrous cobalt molybdate.

The data provided by the TG curves (Figure 3) suggest that the ratio (n) of water molecules per minimum formula is non-stoichiometric, presenting values after heating to 500 °C of 1.03, 1.00, 0.98, and 0.96 for x equal to 0, 0.03, 0.06, and 0.12 of copper in the formation of Co_1−x_Cu_x_MoO_4_·nH_2_O, respectively. Previous works have reported different values of n, which result in the following empirical formulas: CoMoO_4_·0.65H_2_O [25], CoMoO_4_·0.9H_2_O [40], and CoMoO_4_·3/4H_2_O [17]. The TG curves (Figure 3) clearly indicate that doping reduced the number of water molecules of crystallization (related to the first step of dehydration), without significantly changing the number of water molecules that were coordinated to Co atoms (related to second step of dehydration) in the clusters (CoO_6_)_Oh_.

In addition, the presence of copper ions altered the thermal behavior of the CoMoO_4_·nH_2_O after the dehydration process, which is evidenced by the displacement of the TG and DTG curves (Figure 3e–b). The profile of the DTA curves indicated that, after the total loss of water molecules, an exothermic transformation occurred in the structure of Co_1−x_Cu_x_MoO_4_·nH_2_O that is associated with the transition from the hydrated phase to the β phase of cobalt molybdate [15,25,39]. This indicates that the progressive increase in the percentage of copper reduced the energy that is needed to promote dehydration of cobalt molybdate, thus facilitating the transition from H-CoMoO_4_ phase to the β-CoMoO_4_ phase. Another interesting point was observed in the DTA curve of the material Co_0.88_Cu_0.12_MoO_4_·0.96H_2_O, which presented a distinct profile in relation to the other doped samples, indicating that high substitution values for copper atoms led to the existence of more than one thermal event related to the transition from the hydrated phase to the β phase-CoMoO_4_.

### 2.4. Morphological and Elementary Analysis

The images of the samples of pure CoMoO_4_·1.03H_2_O (Figure 4a) and doped with copper (Figure 4b–d) clearly reveal that the adopted co-precipitation method generated particles that consist of well-segregated morphologically uniform micrometric rods with a wide size distribution. A very wide distribution of particle sizes was observed, which ranged from nano to micrometric rods, the average particle length was estimated from the particles in the images in Figure 4, using the program Image J version 6.0 (LOCI, University of Wisconsin, Madison, WI, USA).

The particles of CoMoO_4_·1.03H_2_O exhibited a micrometric rods size format with an average length of 2.08 ± 0.98 µm. Table 2 indicates the morphology and size of the CoMoO_4_·H_2_O crystals that were obtained by the co-precipitation method reported in previous studies [48,49].

The co-precipitation method can obtain samples with different particle sizes and regular morphologies, as can be seen in the data in Table 2. The length of the crystals of CoMoO_4_·1.03H_2_O (2.08 μm) is in the size range that was obtained by Blanco-Gutierrez [15] and Costa [25], together with their respective collaborators, so that the adopted synthesis methodology did not require 12 h of agitation [15] to maturate the precursor mixture or pH correction [25]. The doped materials had an average length of 2.22 ± 1.14, 2.40 ± 1.36, and 2.92 ± 1.39 µm for 3.0, 6.0, and 12% copper, respectively. Therefore, the doping process employed leads to the formation of hydrated cobalt molybdate microrods with a longer average length as the concentration increases of dopant used, in this case copper (Figure 4).

Table 3 presents the elementary composition of the synthesized materials as molar proportions, calculated from the EDS spectra (Figure 4e–h), where the synthesis method via co-precipitation resulted in the desired doping percentages (3.0, 6.0, and 12%), corroborating with the analysis of the FTIR spectra (Figure 2). This indicates a decrease in the relative intensity of the band by 432 cm^−1^, referring to the construction of the CoMoO_4_ groups, due to the increase in the replacement of cobalt ions by copper ions during the process of doping. Table 4 also shows that pure cobalt molybdate is deficient in molybdenum while the presence of copper as a dopant provides richer materials in molybdenum.

### 2.5. Antibacterial Testing of Co_1−x_Cu_x_MoO_4_·nH_2_O Materials

Table 5 lists the values related to Minimum Inhibitory Concentration (MIC) determined for the Co_1−x_Cu_x_MoO_4_·nH_2_O materials.

The data presented in Table 5 indicate that the antibacterial response of CoMoO_4_·1.03H_2_O was more pronounced against *S. aureus*, which showed a 50% reduction in MIC when compared to the system containing *E. coli* strains. This difference may be related to the distinction between the peptidoglycan layers of the bacterial cells of *S. aureus* (gram-positive) and *E. coli* (gram-negative), while the cell wall of *E. coli* consists of a non-continuous envelope that formed by a thin layer of peptidoglycan surrounded by an outer membrane with hydrophilic lipopolysaccharides [49], the wall of *S. aureus* consists of a large layer of peptidoglycan, teichoic acids, and proteins [50].

The negatively charged surface groups (carboxyl or phosphate), which are present in the *E. coli* cell wall, can form bonds with the metal, released by the CoMoO_4_·1.03H_2_O during the process called biosorption, destroying the integrity of the cell membrane and causing protein clotting [51]. It is also known that the coordination and redox chemical properties of cobalt can lead to its non-specific binding to various proteins, displacement of other metals (usually iron) from their natural binding sites, and the generation of free radicals [52]. Previous studies have demonstrated that the toxicity of cobalt to *E. coli* occurs mostly due to its direct competition with iron, mainly affecting the synthesis of Fe–S clusters or indirectly via cobalt-mediated oxidative depletion of the free thiol pool [53,54]. On the other hand, cobalt chloride concentrations ranging from 100 to 400 µM were considered to be adequate for the ideal growth of *E. coli* [55].

In addition to the metal ion binding to biomolecules such as proteins, enzymes, and peptidoglycans present in the cell wall, El-Shaafaf et al. [51] showed that metal can induce bacterial death by another route, in which it intercalates with phosphorus (compositional element is the teichoic acids that make up the cell wall of *S. aureus*) elements in bacterial DNA, preventing the replication and expression of the ribosomal subunit protein and other cellular proteins. Consequently, the data that are presented in Table 5 in relation to the MIC of the analyzed systems infer that the metal performed this route more efficiently when they penetrated the cell wall of *S. aureus*.

The MIC values that were acquired with the copper-doped hydrated cobalt molybdate (Co_1−x_Cu_x_MoO_4_·nH_2_O) were almost equivalent for the gram-positive and gram-negative strains (Table 5). The gram-negative strains of *E. coli* showed an increase in antimicrobial activity of 75% in the MIC value when the percentage of copper was 3% (Co_0.97_Cu_0.03_MoO_4_∙H_2_O). Probably, the higher degree of hydration, which was inversely proportional to the doping percentages of the Co_1−x_Cu_x_MoO_4_·nH_2_O microcrystals (Figure 3), facilitated their passage into the interior of the *E. coli* cell wall, doubling the resistance level of this gram-negative strain to H-CoMoO_4_ doped with 3% copper (Co_0.97_Cu_0.03_MoO_4_·H_2_O) as compared to microcrystals doped with 6% (Co_0.94_Cu_0.06_MoO_4_· 0.98H_2_O) and 12% (Co_0.88_Cu_0.12_MoO_4_ ·0.96H_2_O) of copper.

The mechanism of the antibacterial action of Cu^2+^ ions occurs when they react with biological proteins, thiol (–SH), amino (–NH_2_), and other sulfur-containing functional groups present in the nucleic acid, in order to reduce the activity of these biomolecules, inhibiting bacterial metabolism, according to Zhang et al. [56]. Some of the strains of *E. coli*, which have additional genes encoded by plasmid that confer resistance to this metallic element, can survive in environments rich in copper, and these strains have greater resistance to Cu^2+^ ions than to Cu^1+^ ions [57,58]. Neumann and Leimkühler [59] reported that the activity of molybdoenzymes, such as sulfite oxidase, is inhibited by high concentrations of heavy metals, including copper, in the cell, which interferes with the toxicity of this metal against *E. coli*.

In terms of efficiency, the MIC values that were obtained in the tests conducted with the copper-doped materials increased in the presence of *E. coli* and *S. aureus*, which indicated that CoMoO_4_∙1.03H_2_O exhibited greater antibacterial capacity. In other words, in the presence of Cu, gram-negative (*E. coli*) and gram-positive (S. aureus) bacteria showed greater resistance to H-CoMoO_4_. Recently, Aftab et al. [37] reported that pure NiO behaved as a very good antibacterial agent against *E. coli* (gram-negative) bacteria; however, the presence of 0.1–25% Cu as a dopant gradually decreased its antibacterial capacity. On the other hand, Khan et al. [38] found that doping ZnO with 1 and 5% Cu only exerted antibacterial activity against gram-positive bacteria (*B. subtilis* and *S. aureus*).

The increase in the amount of copper as a dopant that interfered in the molar ratio Co/Mo (Table 4), transforming the hydrated cobalt molybdate into a material, progressively rich in molybdenum, which is a micronutrient involved in the biosynthesis of molybdopterins (a class of cofactors related to a series of physiological processes essential to life [60,61]. Thus, the longer survival of the strains in the presence of the doped material may not be a direct effect of the copper concentration, but rather it is related to the excess molybdenum caused by the doping process, providing greater surface electronegativity to the Co_1−x_Cu_x_MoO_4_·nH_2_O materials.

Among the investigated systems, the one with the best antibacterial performance is made of CoMoO_4_·1.03H_2_O in gram-positive strains of *S. aureus*, which did not show an inhibitory growth process that was enhanced with the presence and/or with the progressive increase of copper as a dopant of the Co_1−x_Cu_x_MoO_4_∙nH_2_O materials. To elucidate the bactericidal effect of Co_1−x_Cu_x_MoO_4_∙nH_2_O materials on the morphology of *S. aureus*, topographic and three-dimensional (3D) images were captured, through Atomic Force Microscopy (AFM), for the control systems and those containing the respective material (Figure 5).

Figure 5a,f show the shape and organization of untreated *S. aureus* cells. The cell wall of bacteria of the *Staphylococcus* genus has a spherical morphology and it is grouped into “clusters” that function as a biological barrier consisting of a phospholipid bilayer with incorporated proteins that are responsible for separating the cell interior from the external environment to maintain a relatively stable internal environment for the occurrence of biochemical reactions [62,63].

The integrity and configuration of the cell wall of the cells observed in the images of the systems treated with Co_1−x_Cu_x_MoO_4_·nH_2_O are deformed (Figure 5b–e,g–j). When exposed to cationic agents, microorganisms suffer the following sequence of events: (i) adsorption and penetration of the agent in the cell wall; (ii) reaction with the cytoplasmic membrane (lipid or protein), followed by membrane disorganization; (iii) leakage of low molecular weight intracellular material; (iv) degradation of proteins and nucleic acids; and, (v) lysis of the wall caused by autolytic enzymes, promoting the loss of structural organization and integrity of the cytoplasmic membrane, together with other harmful effects to the bacterial cell [64].

## 3. Materials and Methods

### 3.1. Reagents

The precursor reagents that were used to obtain the materials CoMoO_4_·nH_2_O and Co_1−x_Cu_x_MoO_4_·nH_2_O were: (NH_4_)_6_Mo_7_O_24_·2H_2_O (Vetec, 99.0%), CoCl_2_·6H_2_O (Dinâmica, 98.0%) and CuNO_3_·2H_2_O (Aldrich, 99.9%). All of the materials were used without prior purification and the water used was ultrapure (Milli-Q IQ 7000; EMD Millipore, Burlington, MA, USA).

### 3.2. Synthesis of CoMoO_4_·nH_2_O and Co_1−x_Cu_x_MoO_4_·nH_2_O Microrods

The co-precipitation synthesis methodology that was adopted in this work was a very simple way to prepare hydrated cobalt molybdate, and it was not necessary to correct the pH, as previously reported [25,26]. The precursors (NH_4_)_6_Mo_7_O_24_·2H_2_O and CoCl_2_·6H_2_O were dissolved in water (250 mmol L^−1^ of Co^2+^) at 80 °C, 750 rpm, and remained under stirring for 2 h. After homogenization, a purple precipitate formed, which was dried at 80 °C for 24 h. Following the same procedure, Cu^2+^ ions from CuNO_3_·2H_2_O were used as a dopant to obtain different compositions based on the following molecular formula: Co_1−x_Cu_x_MoO_4_·nH_2_O, where x = 0; 0.03; 0.06; and, 0.12. Scheme 1 illustrates an illustration of the methodology by co-precipitation to obtain the suspension of CoMoO_4_ (I), then the microcrystals of the materials: CoMoO_4_·nH_2_O (II), Co_0.97_Cu_0.03_MoO_4_·nH_2_O (IV), Co_0.94_Cu_0.06_MoO_4_·nH_2_O (V), and Co_0.88_Cu_0.12_MoO_4_·nH_2_O (VI). The percentage of Cu^2+^ ions affected the coloring of the hydrated cobalt molybdate powder, which was acquired with the doping process to obtain Co_1−x_Cu_x_MoO_4_·nH_2_O (III) suspensions.

### 3.3. Characterization

The crystalline phases of the materials were characterized by X-ray diffractometry (XRD) in a Bruker diffractometer (D8 Advance, 5465 East Cheryl Parkway, Madison WI, 53711-5373, USA) that was operating at 40 kV/30 mA with Cu-Kα radiation (λ = 1.540 Å) and working in continuous mode with a step size of 0.01° 2θ. Data that were suitable for Rietveld refinements were collected over a10° ≤ 2θ ≤ 80° range. In order to obtain the Fourier Transformed Infrared (FTIR, the company, city, state abbrev. if USA, country) spectra, KBr pellets were used and the analyses were performed on the Perkin Elmer model Spectrum 100 equipment, in the range 4000 to 400 cm^−1^ with 4 cm^−1^ resolution and 32 scans per spectrum. Thermal analyses were performed at 25 to 600 °C while using an analyzer (SDT Q600 V20.9 Build 20, the company, city, state abbrev. if USA, country) under an inert atmosphere with a flow of 50 mL min^−1^ and a heating rate of 10 °C min^−1^. The morphological characteristics of the materials were obtained by Scanning Electron Microscopy (SEM) (Inspect S50–FEI, the company, city, state abbrev. if USA, country) with voltage acceleration to 20 kV, coupled to an Energy Dispersive Spectroscopy (EDS) detector, the company, city, state abbrev. if USA, country. The samples were covered with 10 nm gold before being analyzed (Emitech Q150T).

### 3.4. Antibacterial Test

The antibacterial test was performed using the broth micro-dilution method, as recommended by the Clinical Laboratory Standards Institute (CLSI, 2015), in order to determine the Minimum Inhibitory Concentration (MIC). The test was performed in triplicate inside a level II biological safety cabinet using a 96-well microplate. The strains (*Staphylococcus aureus* ATCC 29213 and *Escherichia coli* ATCC 25922) were previously seeded in Petri dishes containing Mueller–Hinton agar and incubated in a bacteriological incubator for 24 h at a temperature of 35.0 ± 2 °C under aerobic conditions. For MIC determination, the bacteria (5 × 10^5^ CFU mL^−1^) were exposed to a serial dilution of ratio 2 of the Co_1−x_Cu_x_MoO_4_·nH_2_O materials in concentrations ranging from 25 to 0.09 mg mL^−1^ under aerobic conditions and constant agitation. Sterility controls of the medium and material as well as control of bacterial growth were performed.

After determining the Minimum Inhibitory Concentration (MIC), an analysis by Atomic Force Microscopy (AFM) was performed to observe the effect of materials Co_1−x_Cu_x_MoO_4_·nH_2_O on *S. aureus* while using a TT-AFM instrument (AFM Workshop, Signal Hill, CA, USA) in vibrating (tapping) mode. After the deposition of 5 mg mL^−1^ of each sample onto clean mica substrate and drying of the sample at room temperature, 8 µm area scans were made. Once dry, representative images of each sample were performed using Tap300-G−10 cantilevers (TedPella Inc., Redding CA, USA), with a resonant frequency of approximately 238.6 kHz. The images were analyzed using Gwyddion software 2.47.

## 4. Conclusions

The synthesis of the materials, which was conducted by the co-precipitation method, resulted in CoMoO_4_·1.03H_2_O microrods with an average length of 2.08 ± 0.98 µm and crystalline structure deficient in molybdenum, while the copper doped structures (Co_1−x_Cu_x_MoO_4_·nH_2_O) gradually became rich in molybdenum as the percentage of copper increased. The doped materials showed an increase in the average length of the microrods, facilitating the transition from the hydrated phase to the β phase, and this transition in the presence of 12% copper occurs through more than one thermal event. However, doping with copper has reduced the potential of CoMoO_4_ to inhibit the growth of the strains of *S. aureus* and *E. coli*, providing increased MIC values. In the absence of copper, CoMoO_4_∙1.03H_2_O showed better bactericidal performance against gram-positive. Therefore, the results indicate the need to control the percentage of copper as a dopant in biocidal materials, such as hydrated cobalt molybdate.

## Data Availability

The data presented in this study are available in this manuscript.

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
