# Peer review of "Insights into the Antimicrobial Activity of Hydrated Cobaltmolybdate Doped with Copper"

_molecules, 2021, doi:10.3390/molecules26051267_

Round 1

Reviewer 1 Report

The authors investigated the effects of copper doping of hydrated cobalt molybdate, synthesized by the co-precipitation method, on antibacterial activity. I suggest the paper for acceptance in Molecules after minor revision (corrections to minor methodological errors and text editing).

1 – The authors should add the uncertainties of the microstructures’ sizes

2 - Scheme 1 and figure 3 have poor quality image

3 – I suggest adding “and elemental” to the title of the sub-section “3.4. Morphological analysis”

4 - Table 3 should have units

5 - In line 353 there is a mistake, figure 6 does not exist, it should be figure 5

Author Response

Dear,

Please see attachment,

Best Regards

Eduardo Triboni

Reviewer 2 Report

  1. The title of this article is “Insights into the antimicrobial activity of hydrated cobalt molybdate doped with copper”. Several experiments of Cu doping concentration have been done in this article. The aim of this paper is to investigate for the effects of copper doping of hydrated cobalt molybdate, synthesized by the co-precipitation method, “antibacterial activity” and to elucidate the structural and morphological changes caused by the dopant in the material. However, the author only shows the results of antibacterial testing without Cu dopant and with 12% copper dopant in section 3.5. The results can not indicate that an increase in the average length of the microrods can caused an increase in MIC values against aureus and E. coli strains. The author must show all of the results of antibacterial testing with various copper dopant concentration of CoMoO4·1.03H2O microrods in Table 5.
  2. The author elucidates the bactericidal effect of CoMoO41.03H2O on the morphology of S. aureus, topographic and 3D images by Atomic Force Microscopy (AFM). The other topographic and 3D images for Co1-xCuxMoO4·nH2O must be shown in this paper to prove the bactericidal effect will be increased by the optimal Cu dopant concentration of Co1-xCuxMoO4·nH2O in Figure 5.
  3. The MTT assay (3-(4,5-Dimethylthiazol-2-yl)-2,5-diphenyltetrazolium bromide) may be done by CoMoO41.03H2O and Co1-xCuxMoO4·nH2O.
  4. This article uses too many words to insert hyphens, which makes it difficult to read. It is recommended to use commercial layout software to reformat this article.

Author Response

Dear,

Please see the attachment,

my Best regards

Eduardo R Triboni

Round 2

Reviewer 2 Report

The author has revised and supplemented the original lacking data in this paper. This article is sufficient to confirm the importance of its experiment.